# TimeScope: Towards Task-Oriented Temporal Grounding In Long Videos

## Abstract

Identifying key moments in long videos is essential for downstream understanding and reasoning tasks. In this paper, we introduce a new problem, Task-oriented Temporal Grounding (**ToTG**), which aims to localize time intervals containing the necessary information based on a task's natural description. Along with the definition, we also present **ToTG-Bench**, a comprehensive benchmark for evaluating the performance on ToTG. ToTG is particularly challenging for traditional approaches due to their limited generalizability and difficulty in handling long videos. To address these challenges, we propose **TimeScope**, a novel framework built upon progressive reasoning. TimeScope first identifies a coarse-grained temporal scope in the long video that likely contains the key moments, and then refines this scope through fine-grained moment partitioning. Additionally, we curate a high-quality dataset, namely **ToTG-Pile**, to enhance TimeScope's ability to perform progressive temporal grounding effectively. Extensive experiments demonstrate that TimeScope consistently outperforms both existing temporal-grounding methods and popular MLLMs across various settings, highlighting its effectiveness in addressing this new challenging problem.

## 1 Introduction

Temporal grounding is crucial for a wide range of applications Gao et al. (2017); Yuan et al. (2021), such as video question-answering and abnormality surveillance. It has been a long-standing research topic, and recent progresses on multi-modal large language models (MLLMs) have substantially advanced the study in this area Zeng et al. (2025); Ren et al. (2024); Wang et al. (2025c); Li et al. (2025e); Huang et al. (2024b); Guo et al. (2025). However, most existing methods focus on temporal grounding based on explicit descriptions of the target. For example, a query like "identify the moment where a little boy is holding a basketball" clearly specifies the event to be localized, allowing models to perform grounding through straightforward visual–text alignment. In contrast, real-world applications often require a more implicit form of temporal grounding. In these scenarios, the model must localize key moments based on a task's natural description, where the relevant information is not directly stated. For instance, given the task "why the boy looks happy when he comes home", the key moment corresponds to the event "he receives a gift from his friend". Although the importance for completing the task, its connection to the task description is indirect, making accurate grounding far more challenging.

In this paper, we introduce a new problem, called **Task-oriented Temporal Grounding** (**ToTG**), to formally conceptualize the aforementioned challenge. Specifically, given a task's natural description, the goal of ToTG is to identify the time intervals within a video that contain the necessary information to solve the task. To facilitate research in this new area, we also create **ToTG-Bench**, a comprehensive benchmark designed to evaluate temporal grounding performance across a diverse set of real-world, long-video understanding scenarios. This benchmark provides a unified and challenging testbed for systematically comparing different approaches and accelerating progress in this emerging area.

The ToTG problem presents significant challenges for existing temporal grounding methods due to two key limitations. First, performing fine-grained grounding in long videos is inherently difficult, as models must sift through vast amounts of content filled with distracting and irrelevant information Li et al. (2025d). Second, current methods often struggle with generalizability, as they are typically

trained to localize moments based on explicit event descriptions Anne Hendricks et al. (2017); Zala et al. (2023); Oncescu et al. (2021), rather than the implicit and diverse natural task descriptions encountered in real-world scenarios.

To address the above challenges, we propose **TimeScope**, a novel framework designed to tackle the ToTG problem through progressive reasoning. TimeScope operates in two stages to accurately localize crucial time intervals in long videos. First, it leverages abstracted video representations that capture comprehensive, high-level information about the entire video while intentionally sacrificing less important details. Using these abstract representations, TimeScope estimates a coarse-grained temporal scope, narrowing down the search to a subspace that are most likely to contain the needed information. Second, TimeScope re-loads detailed video representations for the selected scope and performs fine-grained partitioning to precisely localize the key moments within that region. This progressive process enables the model to effectively handle long videos while achieving high grounding accuracy. In addition to the core framework, we curate a new dataset, **ToTG-Pile**, specifically designed to optimize TimeScope's performance on task-oriented temporal grounding. ToTG-Pile is sourced from diverse, real-world long-video datasets and annotated through a carefully designed pipeline which ensures both quality and diversity. TimeScope is trained on ToTG-Pile using a two-stage supervised fine-tuning strategy. In the first stage, we warm up the model's temporal grounding capability by directly predicting target time intervals from the input video and task description. In the second stage, we establish the progressive reasoning capability by training the model to first estimate coarse-grained temporal scopes using abstracted video representations, and then refine its predictions to identify fine-grained time intervals based on detailed representations.

We conduct comprehensive experiments across a wide range of scenarios, evaluating TimeScope not only on ToTG-Bench but also on popular benchmarks for traditional temporal groundingCaba Heilbron et al. (2015); Sigurdsson et al. (2016); Cheng et al. (2025). The results demonstrate that TimeScope delivers significant improvements over existing approaches, including both dual-encoder-based model, specialized MLLMs for temporal grounding and advanced generic MLLMsBai et al. (2025); Yang et al. (2025a). Further analyses highlight the contributions of each component within our framework, validating its effectiveness in addressing this new and challenging problem. To facilitate future research, we will publicly release all resources, including model, dataset, benchmark, and source code.

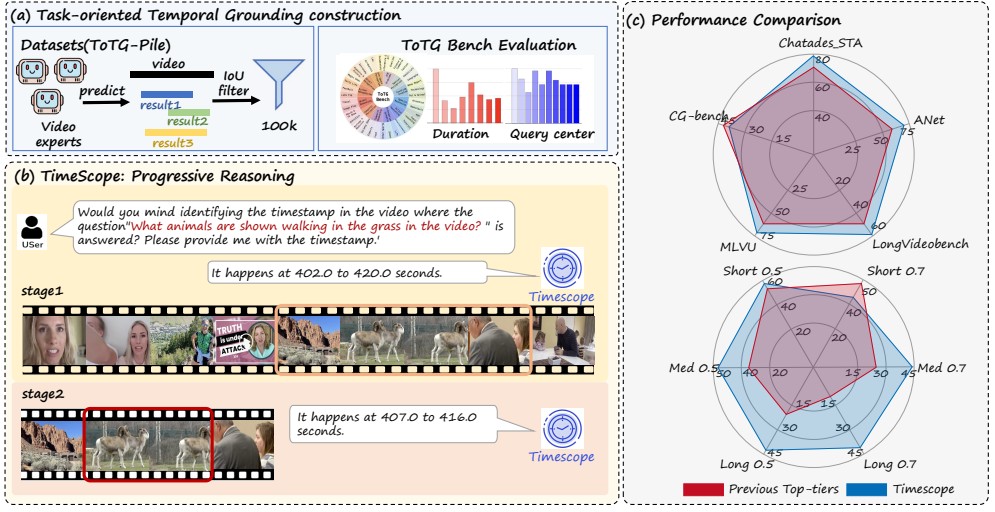

Figure 1: Overview of ToTG-Bench, TimeScope and ToTG-Pile.

## 2 RELATED WORK

### 2.1 VIDEO TEMPORAL GROUNDING

The traditional temporal grounding (TG) task requires models to localize a time interval in a video given a query that explicitly describes the target event. Early approaches are mainly dual-encoder-

based, where video and language features are extracted using different pre-trained encoders (e.g., BERT Devlin et al. (2019), CLIP Radford et al. (2021), SigLip Zhai et al. (2023)), and then fused for time interval prediction Mu et al. (2024); Lei et al. (2021); Moon et al. (2023b;a); Gordeev et al. (2024). These models rely on simple visual-text alignment, which limits their generalization to out-of-domain or more complex queries.

More recently, researchers have explored applying MLLMs to temporal grounding Huang et al. (2024b); Ren et al. (2024); Zeng et al. (2025); Guo et al. (2024); Wang et al. (2025b); Li et al. (2025f). For instance, TimeChat Ren et al. (2024) introduces a time-aware frame encoder that binds visual tokens with their corresponding timestamps at the frame level for temporal grounding. Similarly, TimeSuite Zeng et al. (2025) proposes temporal-adaptive position encoding to strengthen temporal awareness in video representations. Trace Guo et al. (2024) designs a specialized encoder and head for timestamp input, while Time-R1 Wang et al. (2025b) employs a reasoning-guided post-training framework with reinforcement learning and verifiable rewards to improve grounding accuracy. In addition to these specialized MLLMs, recent generic MLLMs (e.g., Qwen2.5-VL Bai et al. (2025), Keye-VL-1.5 Yang et al. (2025b)) have also demonstrated certain capabilities for temporal grounding.

Despite these advances, existing methods are still limited in handling complex queries that require not just locating explicit events but identifying intervals that support completing a task in long video. In particular, most MLLM-based approaches underutilize the generalization potential of MLLMs. Motivated by these limitations, we introduce the new problem of **task-oriented temporal grounding**, along with a benchmark, a dataset, and a dedicated framework to address it.

## 2.2 LONG VIDEO UNDERSTANDING

The field of long video understanding (LVU) has developed rapidly in recent years, with many powerful MLLMs emerging, such as VideoChatFlash Li et al. (2025b), Video-XL-2 Qin et al. (2025a), Eagle2.5 Chen et al. (2025), and InternVL3 Zhu et al. (2025). These models demonstrate strong general video understanding capabilities and serve as versatile backbones for various video tasks, including temporal grounding.

However, precisely localizing or perceiving fine-grained details within second-level intervals remains a major challenge for current LVU MLLMs. To address this, some works introduce additional modules to identify key frames or video segments based on task-oriented queries Wang et al. (2025a); Huang et al. (2025); Yu et al. (2024); Qin et al. (2025b). These modules are typically similarity-based and thus lack deeper semantic understanding of the video content, limiting their compatibility with diverse downstream tasks in long video scenarios.

In contrast, we take a different approach. We post-train LVU MLLMs on our diverse and high-quality task-oriented grounding dataset, and further implement **TimeScope**, a novel framework designed for progressive task-oriented grounding. This enables the model to efficiently and accurately localize critical time intervals in long videos for a wide range of tasks.

## 3 METHOD

In this section, we introduce our new proposed problem, Task-Oriented Temporal Grounding (ToTG) along with its associated benchmark ToTG-Bench, grounding framework TimeScope and training dataset ToTG-Pile in detail. In the following, Section 3.1 formulates the ToTG, which is a more challenging and valuable task compared to traditional temporal grounding. Section 3.2 presents ToTG-Bench, a comprehensive benchmark designed to evaluate model in localizing time intervals based on task description across various long video scenarios. Section 3.3 details our novel framework TimeScope, which aims to address the challenges of TOTG via progressive reasoning. Section 3.4 introduces the dataset TOTG-Piles and its carefully designed pipeline.

## 3.1 PROBLEM FORMULATION

We formally define the Task-oriented Temporal Grounding (ToTG) problem in this section. Given an untrimmed video $V = \{f_1, f_2, \ldots, f_T\}$ and a task-oriented query $q$, the goal is to localize a temporal interval $[t_s, t_e]$ within $V$ that contains the information necessary to accomplish the task described by

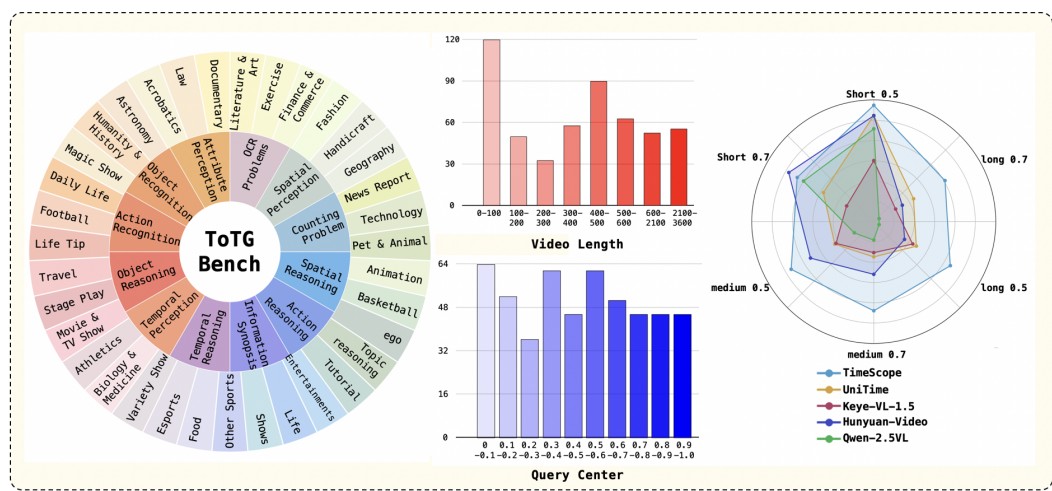

Figure 2: Statistics analysis of ToTG-bench. (Left) Our benchmark covers 12 distinct task types and 35 video categories. (Middle) Video duration and question center distributions. (Right) Performance of various model on ToTG-bench.

$q$. In conventional temporal grounding, the query is usually an event description, including explicit information of the grounding target. This enables the model to locate the corresponding time interval via relatively straightforward visual-text alignment. In contrast, the query in ToTG supplements a higher-level task instruction about the target on this basis. Therefore, the model must locate the significant time interval by comprehensively understanding both visual and textual content, rather than relying solely on surface-level alignment.

## 3.2 BENCHMARKS

To facilitate the evaluation of ToTG, we introduce **ToTG-Bench**, a comprehensive and challenging benchmark. Unlike conventional temporal grounding benchmarks Sigurdsson et al. (2016); Caba Heilbron et al. (2015), ToTG-Bench features queries spanning 12 distinct task types. Its videos are drawn from diverse long-video understanding scenarios and cover durations ranging from a few seconds to nearly one hour. We construct the benchmark through a combination of an effective data filtering pipeline and careful manual annotation, ensuring accurate temporal intervals for each test sample. All of these make ToTG-Bench a high-quality, diverse, and challenging testbed for advancing research in ToTG.

**Benchmark Construction** We first collect samples from four long-video understanding benchmarks (MLVU, Video-MME, LongVideoBench, and V-STaR) as the candidate data for ToTG-Bench construction. The questions in these samples are used as task-oriented queries for grounding. However, not all data are suitable for ToTG, so we design the following filtering pipeline to obtain qualified candidates: 1) **Task type filtering.** We exclude samples whose task types are not compatible with ToTG (e.g., multilingual, summarization, event-ordering tasks and etc.). 2) **Uniqueness filtering.** We retain only samples where the query corresponds to a unique single grounding target. Specifically, we divide each long video into segments and evaluate them using a temporal grounding model followed by manual verification. Samples with exactly only one valid interval contained target are retained. 3) **Information validation.** We further verify that the preserved intervals contain sufficient information to answer the query. We use a advanced MLLM to generate answers based on sampled frames from the predicted intervals, and discard samples for which the answers fail. This pipeline yields a set of qualified candidate data with unique grounding targets, ensuring both annotation quality and fairness in subsequent evaluation. Furthermore, we balance the distribution of candidate data across query task types, video categories, video durations, and temporal positions of the target intervals (We refer to this as *Query Center*) so that each time interval within videos has a roughly equal chance of containing the target. Finally, we carefully manual annotate time interval to ensure the quality of each sample.

**Benchmark Statistics** As illustrated in Figure 2, ToTG-Bench demonstrates significant diversity across task types, video categories, video durations and position of target interval. Specifically, it

includes 12 task types (e.g., action reasoning, OCR perception temporal reasoning and etc.), covering a broad range of video understanding tasks [1]. It further comprises 35 video categories, covering a wide range of real-world video domains. The video durations in ToTG-Bench range from a few minutes to up to one hour. Moreover, the target intervals are balanced to be uniformly distributed along the video timeline, ensuring unbiased evaluation.

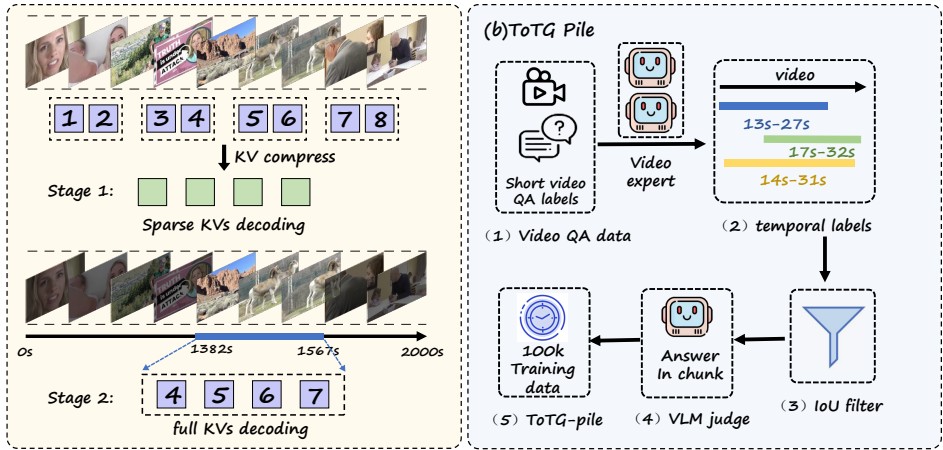

Figure 3: TimeScope framework and ToTG-Pile curation pipeline. **Left:** TimeScope performs progressive temporal grounding, refining coarse windows into precise intervals using fine-grained video features. **Right:** ToTG-Pile pipeline: video experts vote on candidate intervals, and a vision-language model verifies if each interval contains enough information to answer questions.

## 3.3 TIMESCOPE

ToTG requires models to identify the necessary information needed to accomplish diverse tasks in long videos. However, existing temporal grounding models are often distracted by abundant irrelevant content in long contexts, which prevents them from accurately grounding complex task-oriented queries. To overcome this, we design **TimeScope**, a progressive reasoning framework that adapts MLLM backbones for task-oriented temporal grounding. Instead of directly predicting over the full long video at once, TimeScope performs reasoning in two complementary stages, progressively refining from holistic context to precise details (Figure 1).

**Stage 1 (Coarse reasoning).** TimeScope first compresses the cached Key-Value (KV) states of visual tokens (detonated as $KV_{\text{fine}}$) obtained during pre-filling. Through average pooling, the fine-grained $KV_{\text{fine}}$ are distilled into $KV_{\text{coarse}}$, which serves as a compact representation of the global video context. On this compact KV space, the model performs a holistic reasoning step to hypothesize a coarse temporal window $\hat{W}$ that is likely to contain the target.

**Stage 2 (Fine reasoning).** Given the temporal window $\hat{W}$, TimeScope selectively reloads only the fine-grained $KV_{\text{fine}}$ from cache for the corresponding frames, while discarding irrelevant context outside $\hat{W}$. This selective refinement enables the model to reason over detailed visual cues within the localized window, and to output a precise grounding interval.

By framing temporal grounding on long frames as a progressive reasoning process, TimeScope achieves both superior efficiency and accuracy in task-oriented long video temporal grounding.

## 3.4 DATASETS

Existing temporal grounding datasets are mostly limited to short videos with explicit event queries, which makes models trained on them poorly suited for ToTG task. To bridge this gap, we curate

---

[1]We follow the definition of task type and video category from Video-MME

a new large-scale dataset, **ToTG-Pile**, specifically designed for task-oriented temporal grounding. ToTG-Pile is diverse and comprehensive, incorporating both traditional temporal grounding data and newly constructed task-oriented data.

**Traditional temporal grounding data.** Existing datasets for temporal grounding are limited in terms of video duration. To address this, we recaption existing short-video datasets using Qwen2.5-VL to generate concise descriptions. We then concatenate these short videos into longer videos, and the concise descriptions are used to construct explicit temporal grounding queries. Through this pipeline, we obtain a training set of 85k long-video temporal grounding samples, with an average video duration of about 500 seconds (approximately 8 minutes).

**Task-oriented temporal grounding data.** For ToTG data, we collect raw samples from NextQA Xiao et al. (2021), STAR Wu et al. (2024), and CLEVRER Yi et al. (2020). As illustrated in Figure 1(b), the video and answer of each sample are fed into multiple temporal grounding models (refer to *Video Expert*) to generate candidate time intervals containing the answer. A sample is retained only if the Intersection-over-Union (IoU) between intervals predicted by different models exceeds 0.5, and is further evaluated by another MLLM to ensure it contains sufficient information to answer the question. The resulting qualified QA pairs with timestamps are then concatenated into longer videos, yielding training data tailored for task-oriented grounding.

Overall, ToTG-Pile unifies traditional grounding data with newly constructed task-oriented data, ensuring diversity across tasks, durations, and video domains. ToTG-Pile lays the foundation for developing excellent temporal grounding models in ToTG task.

## 4 EXPERIMENT

### 4.1 IMPLEMENTATION DETAILS

We adopt VideoXL-2 as our backbone for two reasons: first, it can ingest very long video sequences, which makes it straightforward to build long-video temporal-understanding methods; second, its internal design interleaves timestamp tokens, giving the model a strong built-in sense of time. TimeScope is trained on ToTG-Pile with a two-stage supervised fine-tuning schedule. In Stage 1 we use the temporal-grounding splits of ToTG-Pile and ask the model to predict the target time interval directly from the raw video and the task description, thereby bootstrapping its basic localization ability. In Stage 2 we apply heavy temporal augmentations—random cropping, shifting, and scaling of the time span—to the training videos, forcing the model to first estimate a coarse temporal window from an abstract video representation and then refine that window into a fine-grained interval using the detailed representation. Throughout training we sample video frames at 1 fps, capping at 300 frames maximum.

### 4.2 BENCHMARKS AND METRICS

For evaluation, we benchmark our method across the following four categories: (i) short-video temporal grounding, including Charades-STA [46] and ActivityNet-Captions; (ii) long-video temporal grounding, including videos longer than 300s from V-STaR and Vid-Chapters-7M; (iii) Video Question Answering (VideoQA), including three general long-video QA benchmarks: CG-Bench, MLVU, and LongVideoBench ; (iv) long video task grounding, including our proposed ToTG-bench. The statistics of the evaluation benchmarks used are listed in Table 5.

Evaluation metrics. For temporal grounding tasks and task grounding tasks, we adopt Recall@1 (R@1) at multiple temporal intersection-over-union (IoU) thresholds and mean IoU (mIoU) as evaluation metrics. Specifically, for temporal grounding benchmarks, we use IoU thresholds of 0.3, 0.5, and 0.7. For general VideoQA tasks, we report standard accuracy metrics.

### 4.3 SOTA PERFORMANCE ON TRADITIONAL TEMPORAL GROUNDING

We conduct a comprehensive comparison of TimeScope against dual-encoder-based methods, MLLM-based methods, and video understanding models on both short- and long-video temporal grounding benchmarks. As shown in Table 1, TimeScope achieves state-of-the-art (SOTA) performance across all benchmarks. For instance, on Charades-STA, TimeScope reached an R1@0.7 score

| Method | Charades-STA | | | ActivityNet | | |
|---|---|---|---|---|---|---|
| | R1@0.5 | R1@0.7 | mIoU | R1@0.5 | R1@0.7 | mIoU |
| **Open-source VLP Method** | | | | | | |
| 2D-TAN Zhang et al. (2020) | 45.8 | 27.9 | – | 60.4 | 43.4 | – |
| UniVTG Lin et al. (2023) | 60.2 | 38.6 | – | 56.1 | 43.4 | – |
| SSRN Zhu et al. (2022) | 65.5 | 42.6 | – | – | 54.5 | – |
| SnAG Mu et al. (2024) | 64.6 | 46.2 | – | – | 48.6 | – |
| EaTR Jang et al. (2023) | 68.4 | 44.9 | – | – | 58.2 | – |
| **Open-source MLLMs Method** | | | | | | |
| TimeChat Ren et al. (2024) | 32.2 | 13.4 | 32.2 | 36.2 | 20.2 | 21.8 |
| VTimeLLM Huang et al. (2024a) | 27.5 | 11.4 | 31.2 | 44.0 | 27.8 | 30.4 |
| VideoChat-Flash Li et al. (2024) | 53.1 | 27.6 | – | – | – | – |
| TRACE Guo et al. (2024) | 61.7 | 41.4 | 41.4 | 37.7 | 24.0 | 39.0 |
| HawkEye Wang et al. (2024) | 58.3 | 28.8 | – | 55.9 | 34.7 | – |
| TimeSuite Zeng et al. (2025) | 67.1 | 43.0 | – | – | – | – |
| Time-R1 Wang et al. (2025b) | 72.2 | 50.1 | – | 58.6 | 39.0 | – |
| DeepVideo-R1-7B Park et al. (2025) | 71.7 | 50.6 | **61.2** | 33.9 | 18.0 | 36.9 |
| VideoChat-R1-7B Li et al. (2025c) | 71.7 | 50.2 | 60.8 | 33.4 | 17.7 | 36.6 |
| TimeZero-7B Wang et al. (2025b) | 60.8 | 35.3 | 58.1 | 39.0 | 21.4 | 40.5 |
| Temporal-RLT-7BLi et al. (2025a) | 67.9 | 44.1 | 57.0 | 38.4 | 20.2 | 39.0 |
| **TimeScope-7B** | **78.9** | **64.0** | 57.5 | **69.6** | **59.0** | **48.0** |

Table 1: Performance comparison on short video temporal grounding tasks including Charades-STA and ActivityNet.

| Model | R1@0.5 | R1@0.7 |
|---|---|---|
| VTimeLLM-7B Huang et al. (2024a) | 4.1 | 1.6 |
| CLIP Radford et al. (2021) | 5.2 | 2.3 |
| M-DETR Kamath et al. (2021) | **27.3** | 17.6 |
| ReVisionLLM Hannan et al. (2024) | 27.4 | 21.8 |
| Qwen2.5-VL-7B Bai et al. (2025) | 0.3 | 0.1 |
| VITAL-7B Zhang et al. (2025) | 25.8 | 19.5 |
| **TimeScope-7B** | 24.3 | **22.1** |

Table 2: Performance comparison on Long video temporal grounding benchmark Vid-Chapters-7M

| Model | R1@0.5 | R1@0.7 |
|---|---|---|
| Qwen2.5-VL-7B | 0.0 | 0.0 |
| UniTime | 0.629 | 0.629 |
| Keye-1.5-VL-8B | 0.639 | 0.491 |
| **TimeScope-7B** | **0.909** | **0.909** |

Table 3: Performance comparison on long video temporal grounding benchmark V-StaR(duration>300)

of 64.0, surpassing VideoChat-Flash (27.6), TimeSuite (43.0), and Time-R1 (50.1). On ActivityNet, it achieved an R1@0.7 score of 59, outperforming HawkEye (34.7) and Time-R1 (39.0).

Particularly noteworthy is that, compared to most other models, TimeScope maintains a smaller gap between R@1@0.5 and R@1@0.7. This indicates that our model is capable of more precise temporal localization. Furthermore, as shown in Table 3, TimeScope achieved an R1@0.7 score of 90.9 on V-STaR, where video duration reaches 300 seconds, significantly surpassing UniTime's 62.9 and Keye-1.5-VL's 49.1, demonstrating its superior performance and strong capability in handling long-video grounding tasks.

### 4.4 SOTA PERFORMANCE ON TASK-ORIENTED TEMPORAL GROUNDING

Besides traditional temporal grounding, we also evaluate TimeScope on ToTG-Bench, comparing it with MLLM-based models and generic MLLMs. As shown in Table **??**, where "has option" indicates that the options are fed into the model along with certain prompt information, TimeScope achieves outstanding and robust performance across different video durations. Specifically, while TimeScope and ARC-Hunyuan-Video-7B Ge et al. (2025) achieve comparable performance on short

| Method | CG-Bench Acc. | MLVU Acc. | LongVideoBench Acc. |
|---|---|---|---|
| Uniform Sample | 33.87 | 60.53 | 54.82 |
| UniVTG Lin et al. (2023) | 34.87 | 62.56 | 54.67 |
| VTimeLLM Huang et al. (2024a) | 34.60 | 59.52 | 54.30 |
| TimeSuite Zeng et al. (2025) | 32.47 | 58.51 | 53.25 |
| UniTime-Full Li et al. (2025f) | **40.30** | 66.50 | 56.47 |
| **TimeScope-7B** | 39.53 | **70.14** | **59.18** |

Table 4: Performance comparison on Long Video Understanding tasks including CG-Bench, MLVU and LongVideoBench.

| Method | S(R1@0.5) | S(R1@0.7) | M(R1@0.5) | M(R1@0.7) | L(R1@0.5) | L(R1@0.7) |
|---|---|---|---|---|---|---|
| | | | no option | | | |
| Qwen-2.5VL-7B | 0.458 | 0.400 | 0.111 | 0.091 | 0.030 | 0.030 |
| ARC-Hunyuan-Video-7B | 0.523 | **0.484** | 0.36 | 0.260 | 0.175 | 0.162 |
| Keye-VL-1.5-8B | 0.521 | 0.333 | 0.313 | 0.261 | 0.221 | 0.124 |
| UniTime | 0.524 | 0.427 | 0.296 | 0.232 | 0.242 | 0.226 |
| **TimeScope-7B** | **0.574** | 0.436 | **0.47** | **0.44** | **0.435** | **0.405** |
| | | | has option | | | |
| Qwen-2.5VL-7B | 0.475 | 0.413 | 0.111 | 0.091 | 0.030 | 0.030 |
| ARC-Hunyuan-Video-7B | 0.158 | 0.148 | 0.111 | 0.090 | 0.124 | 0.092 |
| Keye-1.5-VL-8B | 0.527 | 0.363 | 0.514 | **0.471** | 0.261 | 0.185 |
| UniTime | 0.557 | 0.441 | 0.317 | 0.274 | 0.287 | 0.256 |
| **TimeScope-7B** | **0.683** | **0.515** | **0.52** | 0.44 | **0.467** | **0.426** |

Table 5: Performance comparison on ToTG-bench. "S" refer to "Short", "M" refer to "Medium", "L" refer to "Long". "has option" represents incorporating the options into the prompt.

videos (within a 5-point difference), TimeScope shows clear superiority on medium and long videos, where it outperforms other models by 10 to 20 points. This indicates that our approach can effectively handle videos of various durations, especially long videos, and demonstrates excellent grounding ability for task-oriented queries.

| Model | Early | Middle | Late | Std Dev ($\Delta$) |
|---|---|---|---|---|
| Qwen-2.5-VL Bai et al. (2025) | 20.0 | 13.4 | 4.1 | 8.0 |
| Keye-1.5-VL Yang et al. (2025a) | 50.0 | 33.3 | 31.1 | 10.4 |
| UniTime-Full Yang et al. (2025a) | 50.9 | 28.1 | 44.0 | 11.6 |
| **TimeScope-7B** | 39.3 | 44.8 | 46.0 | **3.5** |

Table 6: Robustness of different models across target interval positions (R1@0.5). TimeScope has lowest Std Dev on three interval distribution.

## 4.5 Improvement for Long-Video QA

As discussed in Section 3.1, the strong performance of TimeScope on the ToTG task suggests that it can help MLLMs capture the critical information in long videos required for question answering. To validate this point, we first use TimeScope and other temporal grounding models to localize time interval. The frames within the predicted intervals are then fed into Qwen2.5-VL-7B for answering. Their results are compared against the default setting, where frames are uniformly sampled without temporal grounding. We evaluate this experiment on three long-video understanding benchmarks: CG-Bench, MLVU, and LongVideoBench. As shown in Table 4a, TimeScope consistently brings significant improvements across all benchmarks, while other temporal grounding models fail to outperform the default uniform-sampling baseline.

| Model | R1@0.3 | R1@0.5 | R1@0.7 |
|-------|--------|--------|--------|
| TimeScopeScope Zeroshot | 89.7 | 85.2 | 73.8 |
| TimeScopeScope Multi-stage | **92.1** | **90.9** | **90.9** |

| Method | Prefill | Decode | Cost |
|--------|---------|--------|------|
| **zero shot** | | | |
| Stage1 | 1400ms | 672ms | 2072ms |
| **Multi-stage** | | | |
| Stage1 | 1400ms | 441ms | |
| Stage2 | 158ms | 357ms | 2356ms |

(a) Multi-stage performance on the V-STaR bench (duration > 300).

(b) Comparison of time consumption between Multi-stage and zero when the number of input tokens reaches 30k.

Figure 4: The ablation study of the multi-stage reasoning proposed by TimeScope

## 4.6 ABLATION STUDIES

**Effectiveness of Progressive Reasoning.** To assess the effectiveness and necessity of progressive reasoning, we conduct an ablation study comparing two settings: progressive reasoning and single-step reasoning. The study is performed on videos longer than 300 seconds from the V-STaR benchmark, with results reported in Table 4a. We observe that progressive reasoning achieves substantial improvements over the single-step baseline. In particular, for longer videos, progressively narrowing the search space proves both effective and necessary for accurate temporal grounding.

**Efficiency of Progressive Reasoning.** In addition to effectiveness, we also evaluate the computational efficiency of TimeScope. As shown in Table 4b, progressive reasoning introduces only marginal additional overhead compared to single-step reasoning, while still delivering substantial accuracy gains. This efficiency largely benefits from our design of adjusting KV sparsity during progressive reasoning. Overall, TimeScope strikes a cost-effective balance between accuracy and computational cost, making it practical for long-video temporal grounding.

**Robustness against Time Bias.** After analyzing the evaluation results on ToTG-Bench, we find that the *Query Center* (the position of the target time interval) introduces a significant bias that strongly influences the performance of temporal grounding models. As shown in Table 6, Qwen2.5-VL and Keye-1.5-VL tend to perform better when the target time interval appears at the beginning of the video, but their performance drops substantially when the target lies in the middle or towards the end. In contrast, our TimeScope model maintains robust performance across different target interval positions, demonstrating its resilience against this bias.

## 5 CONCLUSION

In this work, we define a new task—Task-Oriented Temporal Grounding (ToTG)—and formally conceptualize the aforementioned challenges. To foster research in this emerging area, we introduce ToTG-Bench, a benchmark designed to evaluate temporal grounding performance on diverse, real-world, long-form video-understanding scenarios. To tackle these challenges, we propose TimeScope, a novel framework that solves ToTG through step-by-step reasoning. To strengthen TimeScope, we release ToTG-Pile, a dataset expressly engineered to optimize MLLMs for task-oriented temporal grounding. Harvested from diverse real-world long-video corpora and annotated via a carefully engineered pipeline, ToTG-Pile provides large-scale, high-quality training data. Extensive experiments across a wide spectrum of settings show that TimeScope achieves substantial improvements over existing methods on both traditional benchmarks and ToTG-Bench. We hope this work will stimulate future research on Task-Oriented Temporal Grounding and propel MLLMs toward deeper temporal understanding of video.

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

OVERVIEW OF APPENDIX

- A: **Use of LLMs**
- B: **Benchmark Details**
- C: **Model Arch and Training detail**
- D: **Experimental Settings & Additional Results**
- E: **VideoQA Task Details**
- F: **Limitations & Future Work**
- G: **Broader Impacts Statement**
- H: **Qualitative Results**

## A    USE OF LLMS

In the process of writing this paper, we utilized a Large Language Model (LLM) solely as a tool for polishing the language and enhancing the readability of the manuscript. The LLM was employed to refine the wording, grammar, and coherence of the text, ensuring that the content is clear and well-structured. However, it is important to note that the LLM did not play any significant role in the conception of research ideas or the formulation of the paper's content. All research ideas, methodologies, and conclusions presented in this paper are the original work of the authors. We understand that we bear full responsibility for the contents of this paper, including any text that has been processed by the LLM. We have carefully reviewed and verified all content to ensure its accuracy and originality. Any contributions made by the LLM were limited to language enhancement and did not involve any form of plagiarism or scientific misconduct.

## B    BENCHMARK DETAILS

To evaluate the performance of our model on temporal grounding and VideoQA tasks, we employ the Intersection-over-Union (IoU) metric and its variants. These metrics quantify the alignment between the predicted temporal window $T^{\text{pred}}$ and the ground truth temporal window $T^{\text{gt}}$. The formal definitions are as follows:

### B.1    IOU AND MIOU

The fundamental IoU metric is defined as:

$$\text{IoU} = \frac{|T^{\text{pred}} \cap T^{\text{gt}}|}{|T^{\text{pred}} \cup T^{\text{gt}}|}$$

The mean Intersection-over-Union (mIoU) is calculated as the average IoU across all test samples:

$$\text{mIoU} = \frac{1}{N} \sum_{i=1}^{N} \text{IoU}_i$$

where $N$ is the total number of test samples and $\text{IoU}_i$ is the IoU value for the $i$-th sample.

We also evaluate performance using IoU thresholds, which measure the percentage of predictions exceeding specific IoU values:

- **IoU@0.3**: Percentage of predictions with IoU $\geq 0.3$

$$\text{IoU@0.3} = \frac{1}{N} \sum_{i=1}^{N} \mathbb{I}(\text{IoU}_i \geq 0.3) \times 100\%$$

- **IoU@0.5**: Percentage of predictions with IoU $\geq 0.5$

$$\text{IoU@0.5} = \frac{1}{N} \sum_{i=1}^{N} \mathbb{I}(\text{IoU}_i \geq 0.5) \times 100\%$$

- **IoU@0.7**: Percentage of predictions with IoU $\geq 0.7$

$$\text{IoU@0.7} = \frac{1}{N} \sum_{i=1}^{N} \mathbb{I}(\text{IoU}_i \geq 0.7) \times 100\%$$

Here, $|T^{\text{pred}} \cap T^{\text{gt}}|$ denotes the duration of the overlapping region between the predicted window $T^{\text{pred}}$ and the ground truth window $T^{\text{gt}}$. $|T^{\text{pred}} \cup T^{\text{gt}}|$ represents the duration of their union, while $|T^{\text{pred}}|$ and $|T^{\text{gt}}|$ denote the durations of the predicted and ground truth windows, respectively. The indicator function $\mathbb{I}(\cdot)$ equals 1 when the condition is true and 0 otherwise.

IoU measures the overall alignment between $T^{\text{pred}}$ and $T^{\text{gt}}$, providing a balanced assessment of both precision and recall by considering the overlap relative to their union. The threshold-based metrics (IoU@0.3, IoU@0.5, IoU@0.7) evaluate the model's ability to produce high-quality predictions meeting different precision standards, while mIoU provides an overall average performance measure across all samples.

## B.2 QUERY CENTER ROBUSTNESS

To measure the effectiveness of balanced query centers in the ToTG-benchmark, we evaluated the sensitivity of state-of-the-art (SOTA) models to query centers. As shown in the figure, experiments indicate that most models perform better on test data with query centers positioned earlier rather than later or in the middle. We measured the difference between the best and worst performance of various models when the query center varies. The results show that Timescope maintains its effectiveness regardless of the position of the query center (for instance, the gap is only 35% at iou@0.3, which is significantly better than Qwen2.5vl's 78% and 23.8% higher than UniTime). This further proves the effectiveness of Timescope's training.

| Query Center | | 0-0.1 | 0.1-0.2 | 0.2-0.3 | 0.3-0.4 | 0.4-0.5 | 0.5-0.6 | 0.6-0.7 | 0.7-0.8 | 0.8-0.9 | 0.9-1 | diff |
|---|---|---|---|---|---|---|---|---|---|---|---|---|
| Qwen2.5VL | R@0.3 | 0.417 | 0.368 | 0.615 | 0.222 | 0.250 | 0.158 | 0.278 | 0.235 | 0.133 | 0.154 | 78% |
| | R@0.5 | 0.250 | 0.158 | 0.462 | 0.185 | 0.062 | 0.158 | 0.111 | 0.011 | 0.066 | 0.000 | 97% |
| | R@0.7 | 0.208 | 0.158 | 0.308 | 0.074 | 0.000 | 0.053 | 0.055 | 0.000 | 0.000 | 0.000 | 100% |
| Keye-1.5-VL | R@0.3 | 0.615 | 0.652 | 0.666 | 0.444 | 0.375 | 0.353 | 0.556 | 0.600 | 0.235 | 0.461 | 64% |
| | R@0.5 | 0.538 | 0.521 | 0.400 | 0.379 | 0.186 | 0.294 | 0.444 | 0.466 | 0.176 | 0.307 | 67.2% |
| | R@0.7 | 0.462 | 0.260 | 0.266 | 0.259 | 0.125 | 0.176 | 0.222 | 0.333 | 0.117 | 0.231 | 74.6% |
| Unitime | R@0.3 | 0.669 | 0.465 | 0.633 | 0.364 | 0.371 | 0.216 | 0.550 | 0.567 | 0.371 | 0.567 | 67.7% |
| | R@0.5 | 0.592 | 0.335 | 0.633 | 0.221 | 0.294 | 0.216 | 0.500 | 0.566 | 0.311 | 0.433 | 65.0% |
| | R@0.7 | 0.476 | 0.335 | 0.367 | 0.186 | 0.176 | 0.163 | 0.500 | 0.455 | 0.194 | 0.433 | 67.4% |
| Timescope | R@0.3 | 0.393 | 0.417 | 0.606 | 0.464 | 0.421 | 0.470 | 0.555 | 0.667 | 0.470 | 0.467 | **35.1%** |
| | R@0.5 | 0.393 | 0.417 | 0.606 | 0.357 | 0.368 | 0.353 | 0.505 | 0.556 | 0.412 | 0.400 | **41.7%** |
| | R@0.7 | 0.357 | 0.375 | 0.533 | 0.286 | 0.263 | 0.294 | 0.400 | 0.500 | 0.353 | 0.267 | **50.6%** |

## C MODEL ARCH AND TRAINING DETAIL

Timescope initializes with the base model of VideoXL2 because it can handle very long sequences, facilitating the construction of long video Task-oriented Temporal Grounding tasks. The overall architecture is shown in the figure above. The architecture of Timescope consists of four components: a visual encoder, Dynamic Token Synthesis (DTS), an MLP projector, and a Large Language Model (LLM). For the visual encoder, we use SigLIP to encode visual inputs into dense visual features. The DTS module, located after the visual encoder, is constructed by combining spatial-temporal attention blocks and 3D convolutional layers. It aims to process visual features extracted from four consecutive visual inputs as a single group. This design has been validated in Video-XL-Pro. Next, the architecture applies average pooling to adjacent features to further compress the representation. Then, a two-layer MLP projector processes these pooled features, projecting them into the embedding space of the LLM. For our LLM, we adopt Qwen2.5-7B.

Furthermore, in Timescope, to enhance the model's temporal understanding capability, we prepend explicit timestamp tokens (e.g., Time: 4.0 Second) for every four consecutive frames group across the entire frames sequence. This direct timing information substantially improves the model's temporal awareness. During training, we randomly perform shift operations (e.g., 4s to 1004 seconds) and cut operations (e.g., cutting out 10-20 seconds of a video as a single data point) on the explicit timestamp tokens of the input video as a whole. This training technique greatly enhances Timescope's multi-stage understanding capability and robustness to query positions, as shown in Table **??**.

## D  EXPERIMENTAL SETTINGS & ADDITIONAL RESULTS

We elaborate on the training and inference details of Timescope. The reported hyperparameters cover Stage 1, and 2, as specified in Table 1. For the inference details, we emphasize the particular context length for different benchmarks, as shown in Table 2.

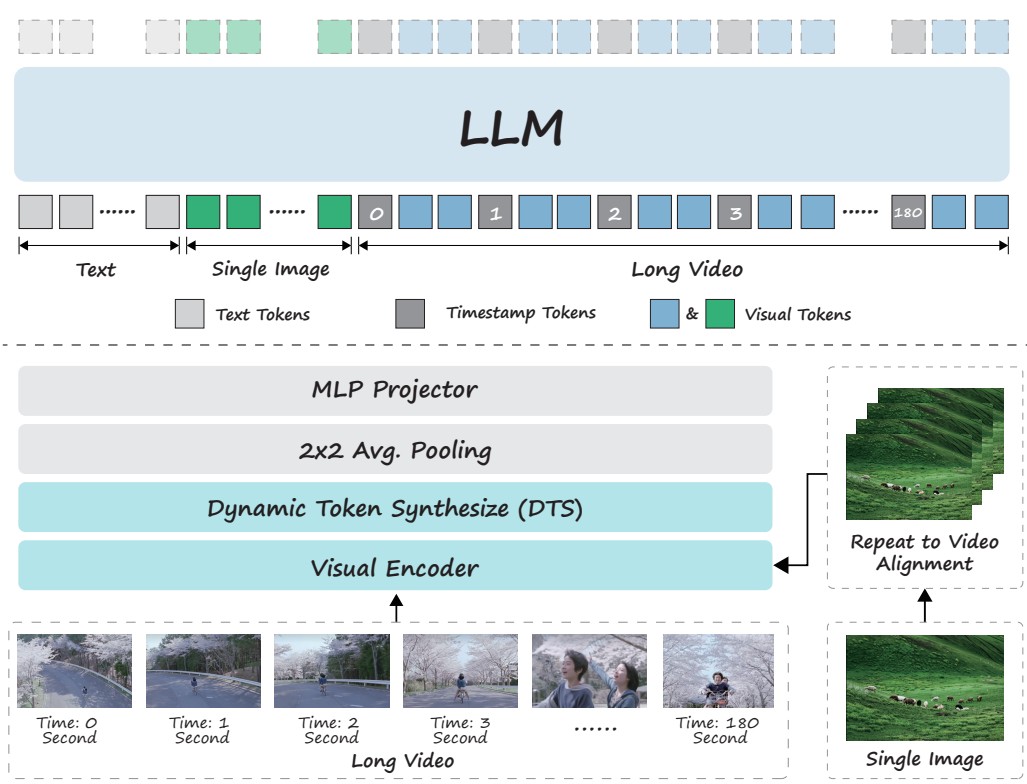

Figure 1: Overview of Timescope.

## E  VIDEOQA TASK DETAILS

### E.1  VIDEOQA WITH TEMPORAL GROUNDING

We use `Qwen2-VL-7B` as the VideoQA model for answer generation. By default, it processes long videos by uniformly sampling 32 frames. However, this sampling strategy may lead to the omission of critical information. To investigate whether temporal grounding models can compensate for this issue, we adopt the following procedure. First, we use different video temporal grounding models to localize the relevant segments for each question. Then, we crop the localized video intervals and input them into `Qwen2-VL-7B`. Specifically, for cropped video segments shorter than 32 seconds, we extend their duration from the center to 32 seconds. Within each interval, we again uniformly sample 32 frames for answer generation.

| Hyperparameter | Stage 1 | Stage 2 |
|---|---|---|
| Overall batch size | 64 | 64 |
| Learning rate | 1e-5 | 1e-5 |
| LR Scheduler | Cosine decay | Cosine decay |
| DeepSpeed ZeRO Stage | ZeRO-2-offload | ZeRO-2-offload |
| Optimizer | Adam | Adam |
| Warmup ratio | 0.3 | 0.3 |
| Epoch | 1 | 1 |
| Weight decay | 0 | 0 |
| Precision | bf16 | bf16 |

Table 1: Hyperparameters of Timescope for different training stages

| Dataset | Context Length |
|---|---|
| Charades-STA | 1fps |
| Activity-Net | 1fps |
| Vid-Chapters | 1fps($<$800 frames) |
| CG-Bench | 1fps($<$800 frames) |
| MLVU | 1fps($<$800 frames) |
| LongVideoBench | 1fps($<$800 frames) |
| V-STaR | 1fps($<$800 frames) |

Table 2: Experimental settings of Timescope.

### E.2 PROMPT TEMPLATE FOR VIDEOQA

We use the same prompt template for all multiple-choice VideoQA benchmarks:

```
System:
You are a helpful assistant.
User:
<video>
Question: <question>
Options:
(A) <Option_A>
(B) <Option_B>
(C) <Option_C>
(D) <Option_D>

Please only give the best option.
Best Option:
Assistant:
```

## F LIMITATIONS & FUTURE WORK

Although Timescope demonstrates exceptional performance on various video temporal grounding and video QA benchmarks, it still has several limitations that warrant further exploration: (i) Timescope is currently constrained to temporal grounding tasks (including traditional temporal grounding tasks and Task-Oriented Temporal Grounding tasks). To enable broader applications in MLLMs, it requires more diverse training data with dense temporal annotations. Incorporating such data into the pretraining process of MLLMs could unlock their potential for handling more temporally complex tasks, such as dense video captioning. (ii) Although Timescope enhances MLLMs with temporal grounding capabilities, relying solely on temporal grounding data limits their reason-

ing and question-answering abilities. The ultimate objective is to develop MLLMs that seamlessly integrate localization, reasoning, and question-answering into a unified framework.

## G  BROADER IMPACTS STATEMENT

Our research introduces a new problem, called Task-oriented Temporal Grounding (ToTG), to formally conceptualize the aforementioned challenge. We have created ToTG-Bench, aimed at evaluating temporal grounding performance in diverse real-world long video understanding scenarios and accelerating progress in this emerging field. This facilitates advancing the complexity of temporal understanding in videos and accelerates the development of models that integrate thinking and traditional temporal grounding capabilities. We hope that ultimately, the two can be integrated, unifying problem thinking and temporal grounding. We have also developed a more accurate and efficient temporal grounding framework, Timescope, to advance the field of long video temporal understanding. This could benefit a wide range of downstream applications, such as anomaly detection, security monitoring, etc. We believe that ToTG and Timescope can advance the development of video temporal understanding.

## H  QUALITATIVE RESULTS

In Figure 2–4, we present the qualitative results of Timescope on ToTG-bench and V-STaR. Timescope demonstrates accurate understanding of questions and the ability to provide temporal grounding. In V-STaR, we show the results when two-stage reasoning is applied, and it can be seen that Timescope exhibits robust performance with good coarse-grained segment retrieval and fine-grained temporal grounding capabilities.

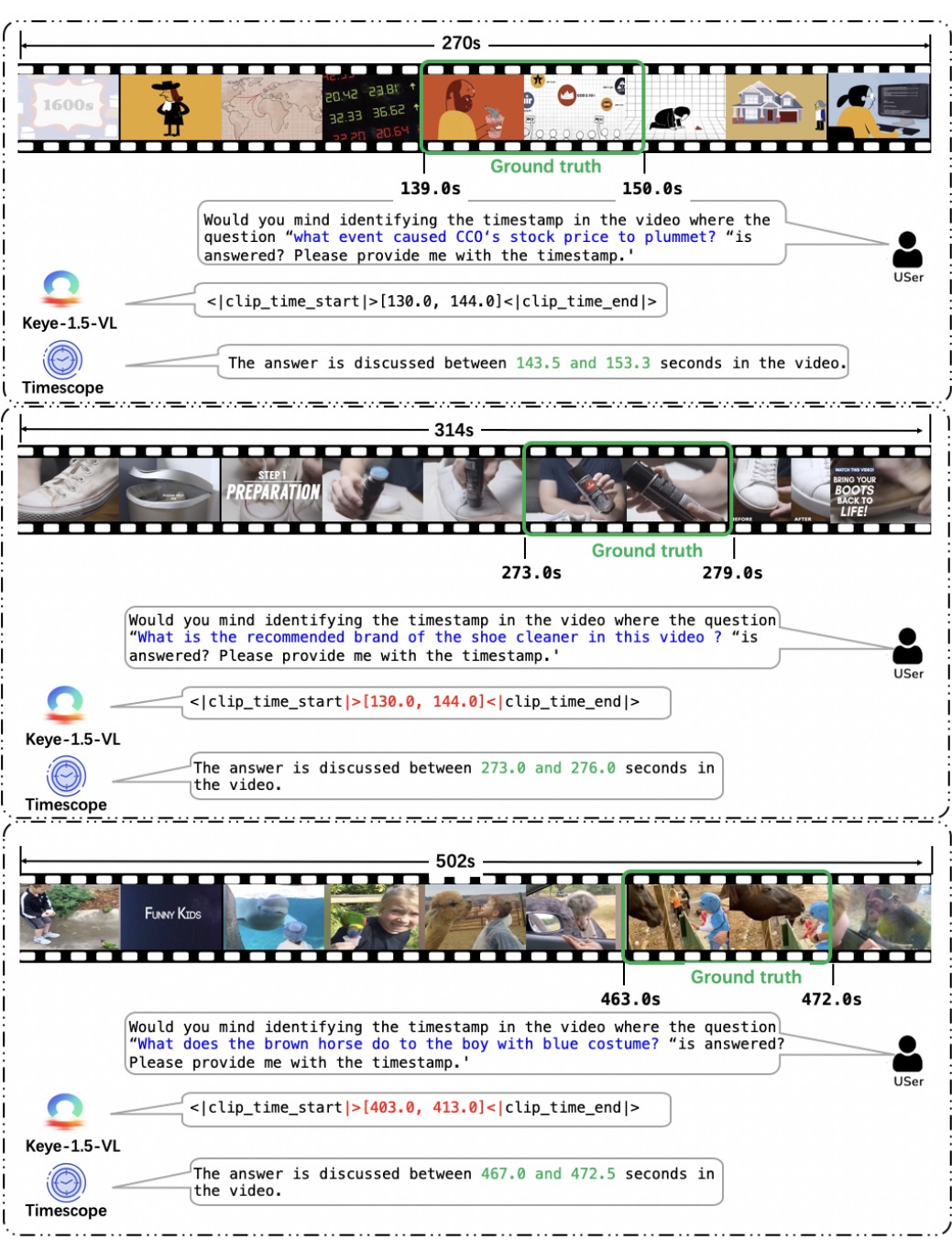

Figure 2: Qualitative Results of Timescope.

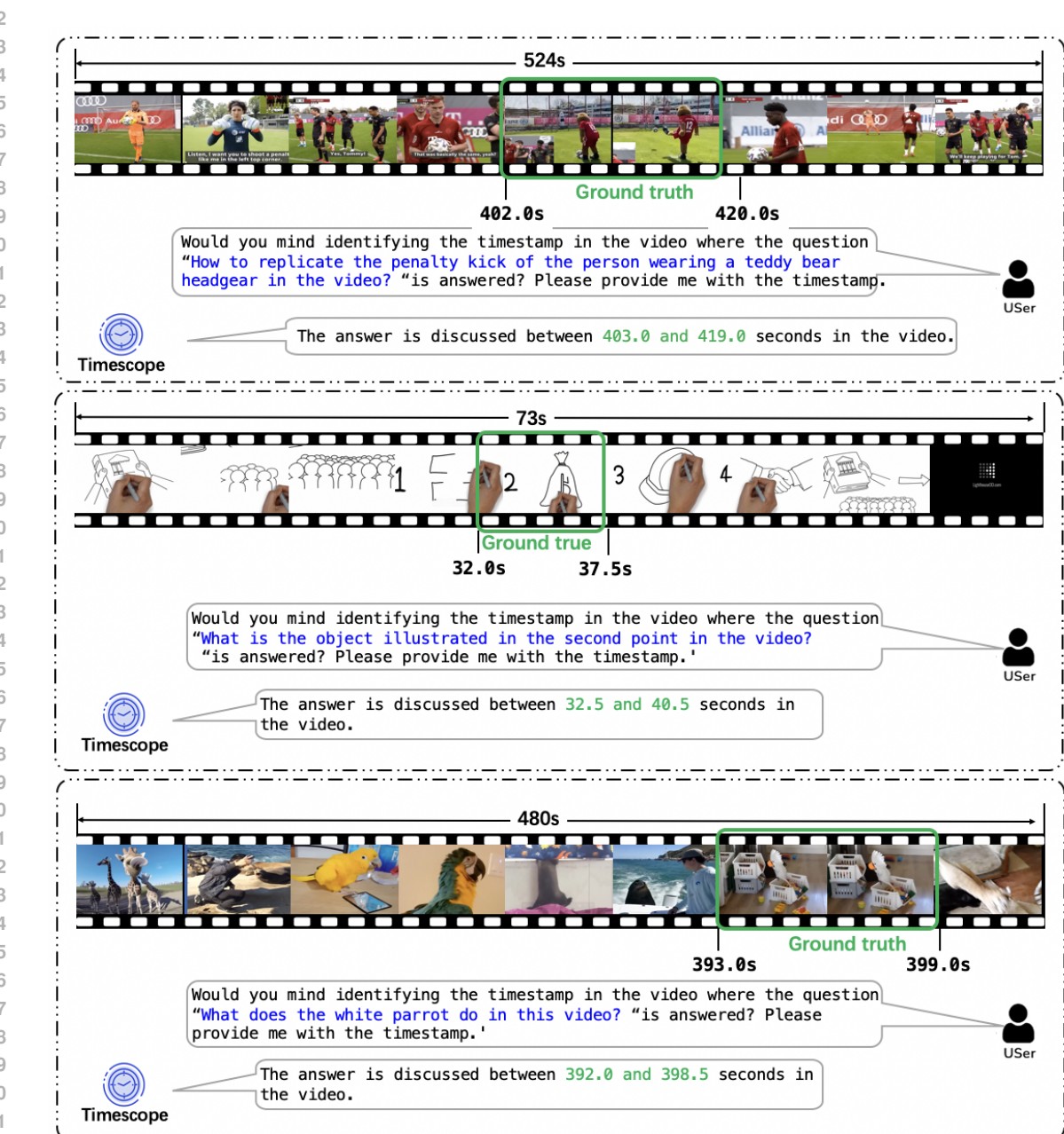

Figure 3: Qualitative Results of Timescope.

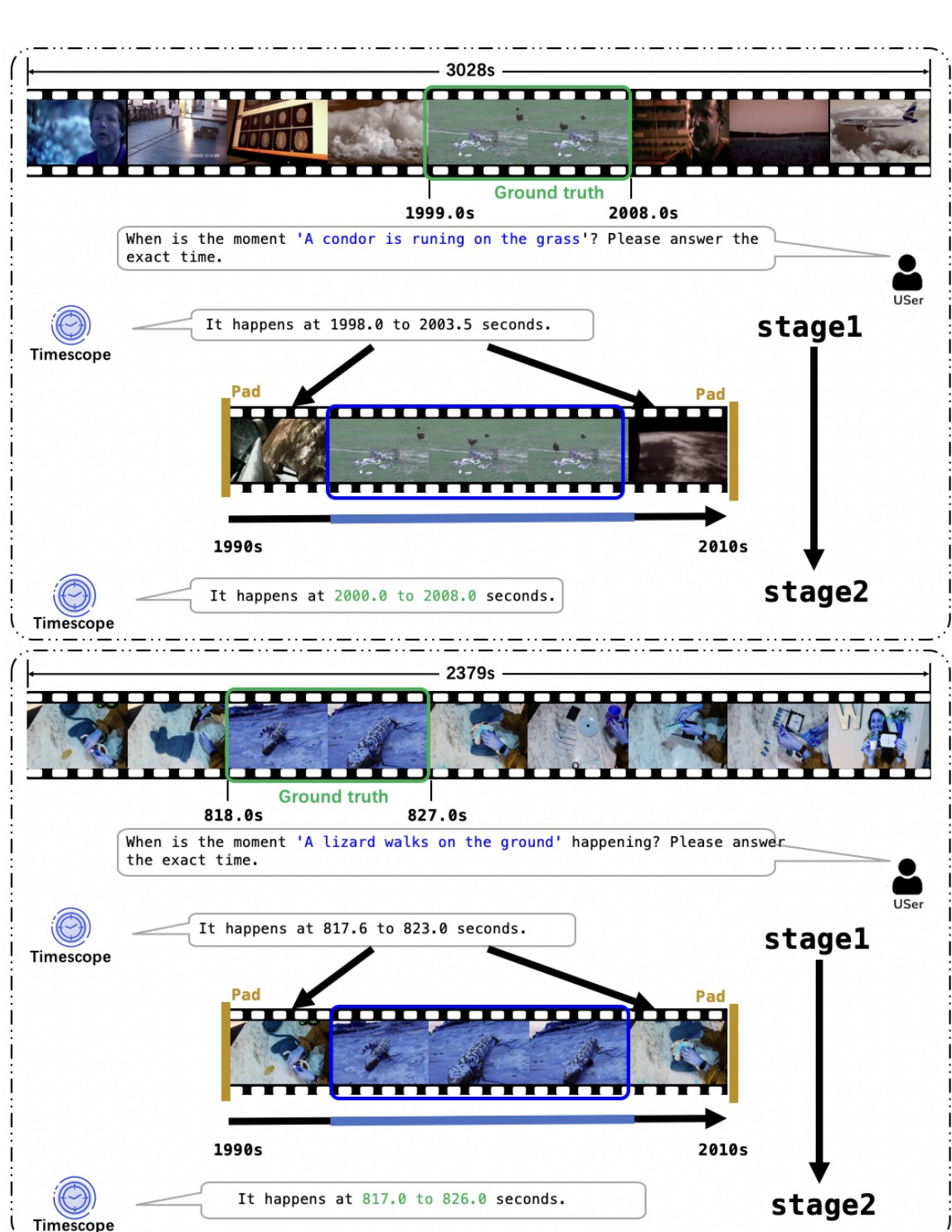

Figure 4: Qualitative Results of Timescope.

