# OpenReview forum: "TimeScope: Towards Task-Oriented Temporal Grounding In Long Videos"
_ICLR.cc/2026/Conference — ICLR 2026 Conference Withdrawn Submission_

### Official Review · Reviewer_MrJC · 2025-11-01

**Soundness:** 3
**Presentation:** 3
**Contribution:** 2
**Rating:** 4
**Confidence:** 4

**Summary:**

This paper introduces Task-oriented Temporal Grounding (ToTG)—localizing the time spans that contain information needed to complete a task described in natural language. The authors contribute ToTG-Bench (evaluation benchmark), ToTG-Pile (training corpus), and TimeScope, a two-stage progressive reasoning framework that first predicts a coarse temporal scope from abstracted video representations and then refines it to precise intervals using fine-grained features. Across benchmarks, TimeScope outperforms strong temporal grounding methods and modern MLLMs, indicating the effectiveness of task-oriented grounding and the proposed coarse-to-fine design.

**Strengths:**

1) **Clear task distinction and motivation.**
   The paper clearly contrasts **explicit event grounding** (e.g., “find the moment a boy holds a basketball”) with **implicit, task-oriented grounding** (e.g., “why does the boy look happy?”), highlighting a practical gap that existing TG benchmarks don’t address.

2) **Thorough benchmark curation.**
   **ToTG-Bench** is carefully constructed with (i) **12 task types** and **35 video categories**, (ii) balanced coverage over video **durations** and **target positions**, and (iii) a **multi-stage filtering** pipeline to ensure quality and diversity.

3) **Strong empirical performance.**
   **TimeScope** delivers sizable gains: **64.0% R1@0.7 on Charades-STA** (vs. **50.1%** for Time-R1), **90.9% R1@0.7 on V-STaR** for **>300s** videos, and **consistent improvements** across both traditional TG and task-oriented benchmarks, validating the coarse-to-fine design.

**Weaknesses:**

1) **Overstated task distinction.**
   The claimed gap between ToTG and traditional temporal grounding is **overemphasized**. Existing video QA benchmarks—**NExT-QA, STAR, CLEVRER** (which the authors themselves use for ToTG-Pile)—already require **localizing temporal evidence** based on questions rather than explicit event phrases. The paper should more rigorously articulate what ToTG adds beyond these settings (e.g., distinct reasoning types, interval properties, or evaluation protocols that materially change model behavior).

2) **Limited technical novelty and unclear task specificity.**
   - **Coarse-to-fine** strategies are extensively studied in video understanding, temporal action localization, and detection. The proposed **KV compression → coarse scope → KV reload → fine prediction** is a **straightforward adaptation** of known practices.
   - The paper does not convincingly show **why this particular design is uniquely suited to “task-oriented” grounding** versus standard grounding. Evidence such as analyses where ToTG demands qualitatively different reasoning, ablations showing disproportionate gains *only* on task-oriented cases, or failures of comparable coarse-to-fine baselines on traditional TG would strengthen the claim.

**Questions:**

Please refer to the Weaknesses.

---

### Official Review · Reviewer_z4zs · 2025-11-02

**Soundness:** 3
**Presentation:** 2
**Contribution:** 2
**Rating:** 4
**Confidence:** 4

**Summary:**

The paper introduces Task-oriented Temporal Grounding, where a model should localize video intervals that contain the necessary information to complete a task expressed in natural language, rather than matching an explicit event description as in previous work. To support this setting, the authors build a benchmark (ToTG-Bench) and a training dataset (ToTG-Pile). They further propose TimeScope, a two-stage coarse-to-fine localization framework: (1) estimate a coarse interval using compressed KV representations, (2) refine to precise boundaries by reloading fine-grained KVs only for the selected segment. Across several grounding and long-video QA benchmarks, TimeScope reports stronger results than the baseline approach.

**Strengths:**

1. **Insightful analysis.** The study in Table 6 and Appendix B.2  highlights an early-center bias in existing methods, which is valuable for diagnosing model behavior.
2. **Strong performance.** TimeScope achieves competitive or superior results on multiple temporal grounding benchmarks and shows that improved grounding can positively impact video-QA performance.
3. **Practical design for long videos.** The coarse-to-fine, multi-stage strategy is intuitive and efficient, making it well-suited for processing long-context videos.

**Weaknesses:**

1. **Positioning of ToTG-Bench vs prior benchmarks.** The paper should more clearly discuss how ToTG differs from existing settings, such as CG-Bench, NExT-GQA (grounding-QA), and ReXTime (question + options). What is uniquely evaluated by ToTG, and why can’t current benchmarks capture this?
2. **Dataset construction details are missing a lot.** Please specify: the size of ToTG-Bench; whether intervals were annotated by MLLMs or humans; any verification/review process; the exact prompts and model used during filtering/curation; data sources for ToTG-Pile and the proportion drawn from traditional TG data. These are important for assessing quality and potential biases.
3. **Handling multi-interval evidence.** Some questions (e.g., in NExT-GQA) require multiple disjoint moments. TimeScope appears to output a single interval. The author can clarify whether the task definition or model can support multi-segment evidence, and report performance.

**Minor**
1. Typos. Line 254: “detonated” → denoted.
2. Missing References in Line 375 and Line 817.

**Questions:**

Please check the three points mentioned in the above weakness section.

Moreover, the reviewer would also raise concerns about the fairness of the comparison. Many baselines in Table 1 use Qwen2.5-VL, while TimeScope uses VideoXL-2. To address fairness concerns, the author may consider reporting results with the same backbone.

---

### Official Review · Reviewer_BQYf · 2025-11-03

**Soundness:** 3
**Presentation:** 3
**Contribution:** 3
**Rating:** 6
**Confidence:** 4

**Summary:**

This paper introduces a new task called Task-Oriented Temporal Grounding (ToTG), which aims to localize time intervals in long videos that contain the information necessary to accomplish a given task described in natural language. To address this, the authors propose TimeScope, a progressive reasoning framework that first performs coarse-grained localization and then refines the prediction to identify fine-grained intervals. Additionally, the authors curate a large-scale dataset, ToTG-Pile, and construct a benchmark, ToTG-Bench, to facilitate evaluation on diverse long-video understanding tasks.

**Strengths:**

1. Novel problem definition: The paper formalizes the Task-Oriented Temporal Grounding task, extending traditional temporal grounding to more realistic scenarios.
2. Dataset contribution: ToTG-Pile enriches the research landscape with diverse, high-quality data, while ToTG-Bench establishes a standardized evaluation setting.
3. Strong empirical results: TimeScope achieves consistent improvements over state-of-the-art models on both traditional and newly proposed benchmarks.

**Weaknesses:**

1. **Unclear distinction between VTG and ToTG** – The paper would benefit from a more explicit explanation of how *Task-Oriented Temporal Grounding (ToTG)* differs from *traditional Video Temporal Grounding (VTG)*. Specifically, the motivation, challenges, and intended targets of ToTG should be more clearly articulated. For instance, while VTG focuses on locating explicit moments described in natural language (e.g., “the person opens the door”), ToTG aims to identify time intervals containing the information necessary to *perform or understand a task* (e.g., “how to cook pasta”). Emphasizing this conceptual difference would help clarify the paper’s novelty and the unique reasoning requirements of ToTG.
2. **Insufficient dataset details** – Although the paper introduces a new dataset for ToTG, the description lacks sufficient detail and illustrative examples. A more thorough presentation of dataset composition—such as the number of videos, duration statistics, annotation procedure, and representative query examples—would improve clarity and reproducibility. Including a few qualitative examples (e.g., input query, ground-truth interval, and reasoning steps) would be highly beneficial.
3. **Missing pseudo code** – The paper references algorithmic steps for TimeScope’s progressive reasoning process but does not include the pseudo code in the main text or appendix. Providing pseudo code would enhance transparency and make it easier for readers to understand and replicate the proposed pipeline.
4. **Limited generalization beyond temporal grounding** – Since TimeScope is primarily trained and evaluated on temporal grounding data, its generalizability to other video-language understanding tasks remains uncertain. The framework’s adaptability to tasks such as *dense video captioning* or *causal reasoning* has not been demonstrated, which slightly limits its broader impact.

**Questions:**

Please refer to the points listed under Weaknesses.

---

### Official Review · Reviewer_uLAw · 2025-11-11

**Soundness:** 2
**Presentation:** 1
**Contribution:** 2
**Rating:** 2
**Confidence:** 4

**Summary:**

This paper introduces a novel task setting, called Task‐Oriented Temporal Grounding (ToTG), in which a long video and a natural language description of a task are given, and the goal is to localize the time interval in the video that contains the information needed to perform or answer that task. For this task, a large benchmark, ToTG-Bench and ToTG-Pile, is presented, covering many video categories, long durations, and task‐oriented queries. In addition, the paper proposes a model framework (i.e., TimeScope) that uses a progressive reasoning or coarse-to‐fine temporal localization strategy: first identify a coarse interval in the long video likely to contain the answer, then refine it to a fine boundary. The model uses caching of key, value (KV) representations, average pooling to compress into coarse representations, followed by reloading fine KV within the hypothesized interval for refinement. Extensive experiments show that TimeScope outperforms existing temporal grounding methods, including several baselines using video‐language models and multimodal LLMs, on both short and long video benchmarks and on the new ToTG benchmark.

**Strengths:**

**[S1]** The motivation is reasonably clear.

**[S2]** The proposed coarse‐to‐fine TimeScope framework is intuitive and aligned with human‐like reasoning.

**[S3]** Building a new benchmark and performing experiments on both standard benchmarks and their proposed new setup are worth admitting.

**Weaknesses:**

**[W1]** Novelty
- Coarse‐to‐fine is interesting, yet an established paradigm. The specific innovations (e.g., caching KV, pooling to coarse, then refine) are incremental and may not generalize or offer new insights. It would help to better situate the method against similar prior work in temporal grounding and long‐video methods.

**[W2]** Benchmark
- The definition of the task ToTG is somewhat vague: how different is it really from “temporal moment retrieval” or “temporal grounding for video QA”? The paper should clarify exactly what makes the task new (e.g., indirect/implicit queries, task‐oriented rather than event‐oriented) and how that adds new challenges.
- In uniqueness filtering, videos containing multiple grounding targets are excluded. As the benchmark provides long videos, multiple grounding targets would be natural.
- Many details are not clear. e.g. how distractor content is handled, how annotation consistency is measured, how inter‐annotator agreement is computed. This makes it hard to assess dataset validity.

**[W3]** Method
- The coarse‐to‐fine strategy assumes that one coarse window can reliably capture the target interval. However, there is limited discussion of failure cases (e.g., multiple dispersed relevant intervals or tasks requiring discontiguous evidence).

**[W4]** Writing
- There are multiple typographical/formatting issues: R@1@0.5, R@1@0.7, spaces before punctuation, variable formatting in tables/figures.
- ‘citep{}’ should be used to refer to papers.

**Questions:**

Please see the weaknesses.

---

### Note · Authors · 2025-11-14

I have read and agree with the venue's withdrawal policy on behalf of myself and my co-authors.